

# Intraoperative hypotension and postoperative pneumonia in patients after selective intracranial tumor resection: a retrospective cohort study

Yuan Chang, Yanqiong Wang, Yinyan Zhou, Huamei Chen, Yuanhua Li, Ruhua Long and Jianlin Shao

Department of Anesthesiology, First Affiliated Hospital of Kunming Medical University, Kunming, Yunnan, China

## ABSTRACT

**Background.** Postoperative pneumonia is common and associated with increased postoperative mortality. Intraoperative hypotension is suggested to be associated with an increased risk of postoperative surgical infection. We aim to explore whether intraoperative hypotension could increase the risk of postoperative pneumonia in adult patients undergoing intracranial tumor resection.

**Methods.** A total of 341 patients who received selective intracranial tumor resection under general anesthesia between January 2018, and December 2022 in a single university hospital were reviewed. Univariate and multivariate analyses were performed. The outcomes included the incidence of postoperative pneumonia and the association between intraoperative hypotension and postoperative pneumonia.

**Results.** The incidence of postoperative pneumonia during hospitalization after intracranial tumor resection was 16.8%. Univariate analysis revealed a history of preoperative smoking, intraoperative mean arterial pressure (MAP) < 55 mmHg, American Society of Anesthesiologists classification (ASA) > 2, the duration of surgery > 4 hours, and the duration of controlled ventilation > 4 hours were identified as possible risk factors ($P < 0.1$). Multivariate analysis revealed a history of preoperative smoking (adjusted odds ratio: 5.205, 95% confidence interval [1.826–14.836], $P = 0.003$) and intraoperative MAP < 55 mmHg (adjusted odds ratio: 3.082, 95% confidence interval [1.447–6.432], $P = 0.003$) were independently associated with postoperative pneumonia.

**Conclusions.** Intraoperative hypotension may be associated with postoperative pneumonia in patients who received selective intracranial tumor resection under general anesthesia.

## INTRODUCTION

Annually, an estimated five million patients globally undergo neurosurgery (*Dewan et al., 2018*). As one of the most prevalent complications following neurosurgery, postoperative pneumonia prolongs hospitalization and increases the risk of postoperative mortality (*Chastre & Fagon, 2002*; *Dasenbrock et al., 2015*; *Kazaure et al., 2014*; *Warren et*

Corresponding author
Yuan Chang, changyuan@ydyy.cn

*al., 2003*). Previous research has identified several patient-specific and surgery-related factors that are closely associated with the occurrence of postoperative pneumonia, such as preoperative anemia, hypoalbuminemia, tumor types, American Society of Anesthesiologists classification (ASA), chronic obstructive pulmonary disease (COPD), smoking history, surgery duration, and mechanical ventilation (*Chughtai et al., 2017*; *Garibaldi et al., 1981*; *Hooda et al., 2019*; *Ryan et al., 2013*; *Shin et al., 2020*; *Turan et al., 2011*). However, these factors are often difficult to optimize before surgery.

Recent studies observed that intraoperative hypotension increased postoperative cardiovascular complications and mortality (*Gregory et al., 2021*; *Huang et al., 2020*; *Kouz et al., 2020*; *Lizano-Diez et al., 2022*). At the same time, intraoperative hypotension also elevated the risk of postoperative infections (*Ishikawa et al., 2014*; *Tassoudis et al., 2011*; *Yamamoto et al., 2007*; *Zhang et al., 2021*). There is limited research investigating the association between intraoperative hypotension and postoperative pneumonia in patients undergoing neurosurgery. As intraoperative hypotension could be aggressively addressed during surgery, clarifying the relationship between intraoperative hypotension and postoperative pneumonia may help to develop evidence-based intraoperative blood pressure management strategies to improve patient outcomes.

Intracranial tumor resection is one of the most common neurosurgeries. The present single-center retrospective study aimed to explore the association between intraoperative hypotension and postoperative pneumonia in adult patients undergoing intracranial tumor resection.

## MATERIALS AND METHODS

### Study design

This study was a retrospective cohort analysis of adult patients admitted to the post-anesthesia care unit (PACU) after intracranial tumor resection under general anesthesia at a single tertiary medical center in China. The Research Ethical Committee of the First Affiliated Hospital of Kunming Medical University approved the study, and the need for written informed consent was waived (2024-L-188, 2024-07-25). Ethical principles, in line with the 1964 Declaration of Helsinki and its subsequent amendments, were strictly followed. The STROBE checklist was provided in the supplementary materials.

### Study population

We collected the electronic medical records of patients aged >18 years who were admitted to the PACU after elective intracranial tumor resection under general anesthesia between January 1, 2018, and December 31, 2022. Before accessing the data, the following exclusion criteria were used: (1) patients undergoing pituitary tumor surgery; (2) patients admitted to the intensive care unit (ICU) after initial surgery during hospitalization; (3) patients were diagnosed pulmonary pneumonia or respiratory failure within 60 days before surgery; (3) pregnant or lactating patients; (4) patients with a history of tracheostomy; (5) patients who have already undergone endotracheal intubation before anesthesia; and (6) patients without invasive arterial pressure monitoring during surgery.

## Anesthesia management

Total intravenous anesthesia or combined intravenous-inhalation anesthesia, anesthetics, fluid admission, and intraoperative ventilation strategy were determined by attending anesthesiologists. According to the protocol of our institution, a 50% concentration of inhaled oxygen was applied during the surgery. Our institution did not designate a standard threshold of intraoperative hypotension to guide perioperative blood pressure management. Planned postoperative admission to ICU was determined by the surgical team the day before surgery. At our institution, patients who exhibit hemodynamic instability, require ongoing vasopressor or inotropic support, need postoperative mechanical ventilation or respiratory assistance, or have undergone extensive or complex neurosurgical procedures are transferred to the ICU following surgery. Unplanned postoperative admission to ICU was determined by the surgical team and anesthesiologists at the end of surgery.

## Data collection

We obtained patient information from the electronic medical record system and the anesthesia information system of the First Affiliated Hospital of Kunming Medical University. Baseline characteristics included age, sex, body mass index, smoking within 180 days before the surgical day, ASA classification, the last value of preoperative hemoglobin and albumin levels, and preoperative comorbidities including hypertension, diabetes mellitus, COPD, coronary artery disease, and stroke. Preoperative baseline blood pressure was the last blood pressure value documented by ward nurses before surgery. Preoperative hypoproteinemia was diagnosed when the last concentration of serum albumin before surgery was less than 35 g/L, and preoperative anemia was diagnosed when the last concentration of hemoglobin before surgery was less than 120 g/L according to the diagnosis protocol of our institution.

We collected the following surgical and anesthetic data from the anesthesia information system: types of anesthesia (inhaled or intravenous anesthesia), types of intracranial tumor, duration of surgery, duration of controlled ventilation, intraoperative body position, volume of minute ventilation, amount of fluid, blood transfusion, estimated blood loss, and administration of neostigmine during emergence. The duration of controlled ventilation was calculated from the time of tracheal intubation to extubation.

Intraoperative blood pressure was automatically collected every 1 min. We applied the following algorithm to identify potential error blood pressure values: systolic arteria pressure (SAP) > 300 mmHg or SAP < 40 mmHg; diastolic arterial pressure (DAP) > 150 mmHg or DAP < 30 mmHg; pulse pressure < 10 mmHg. Error blood pressure values were replaced by the previous values (*Hirsch et al., 2015*). We applied the following formula to calculate the MAP: $MAp = DBP + 1/3 \times (SBP - DBP)$ (*Monk et al., 2015*). Previous study observed that in noncardiac patients, intraoperative MAP < 55 mmHg significantly increases the risk of postoperative cardiovascular complications, regardless of the duration of exposure (*Wesselink et al., 2018*). Therefore, present study defined MAP < 55 mmHg as the threshold for intraoperative hypotension.

## Outcomes

The primary outcome was the incidence of postoperative pneumonia during hospitalization after intracranial tumor resection. The diagnosis of postoperative pneumonia was made collaboratively by ward neurosurgeons and respiratory physicians. The diagnostic criteria for pulmonary pneumonia were as follows: new or progressive infiltrates, consolidation, or ground glass shadows were observed in chest X-ray or computed tomography (CT) examination with at least two of the following three clinical symptoms present simultaneously: (a) fever with a body temperature exceeding 38 °C, (b) presence of purulent sputum or repeated cough, and (c) peripheral blood leukocyte count exceeding $10 \times 10^9$/L or falling below $4 \times 10^9$/L (*Shi et al., 2019*). Chest X-rays or CT scans are only conducted if the patient exhibited one or more of the previously mentioned three symptoms.

## Statistical analysis

Categorical variables are summarized as frequency (%). The incidence of postoperative pneumonia after intracranial tumor resection is presented as point estimates with a 95% confidence interval (CI). To assess the potential risk factors, logistic regression analysis was conducted using 11 variables. Odds ratios (ORs) and their corresponding 95% confidence intervals (CIs) were calculated for the factors associated ($P < 0.1$) during the univariate logistic regression. Subsequently, a multivariate logistic regression was performed using the variables identified in the previous step. All statistical tests were two-sided, and a significance level of $P < 0.05$ was considered statistically significant (Version 26.0.0.0; IBM Corp.).

## RESULTS

A total of 341 patients underwent elective intracranial tumor resection between between January 1, 2018, and December 31, 2022. After data review, 85 patients were excluded from the study, including 73 who were admitted to the intensive care unit during hospitalization and 12 who required reoperation during the initial hospitalization. Consequently, 256 patients were included and considered for the final analysis (Fig. 1).

Of the 256 patients included in the analysis, 43 (16.8%, 95% CI [12.3%–21.3%]) experienced postoperative pneumonia. Table 1 presents the comparison of baseline parameters between patients with postoperative pneumonia and patients without pneumonia. Compared with patients without postoperative pneumonia, patients with postoperative pneumonia were more likely to have a history of preoperative smoking.

Table 2 presents the comparison of perioperative parameters between patients with postoperative pneumonia and patients without pneumonia. Patients with postoperative pneumonia experienced longer durations of intraoperative hypotension, surgery, and controlled ventilation in comparison to patients without postoperative pneumonia.

### Primary analysis

After univariate analyses, a history of preoperative smoking, intraoperative MAP < 55 mmHg, ASA classification, duration of surgery > 4 h, and duration of controlled

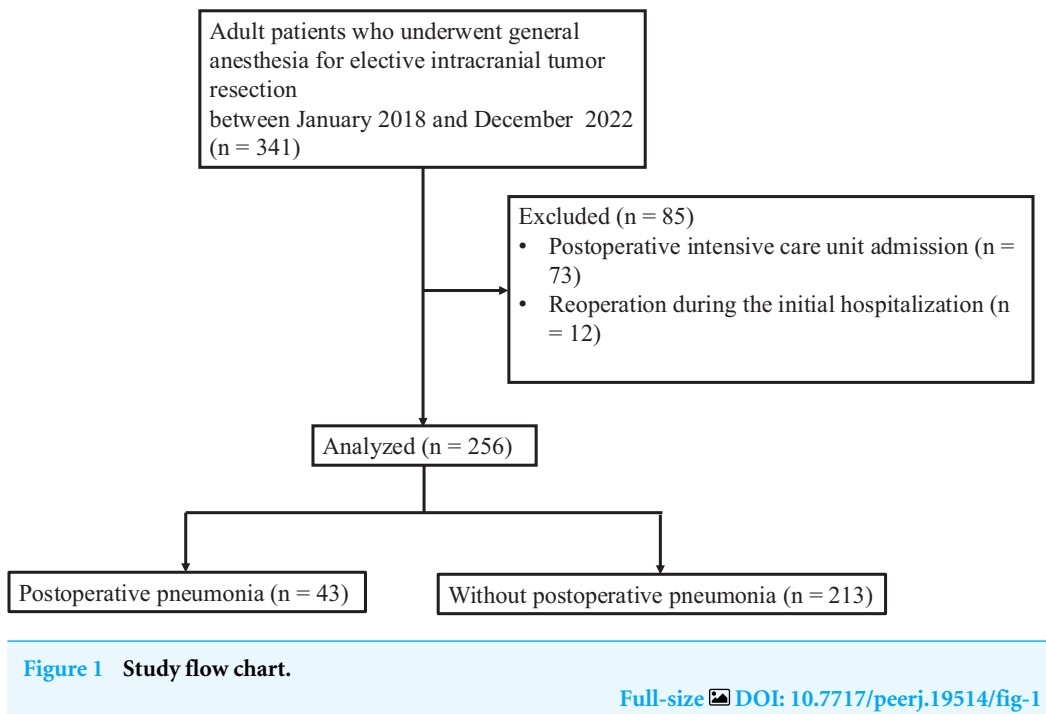

**Figure 1  Study flow chart.**

ventilation > 4 h were identified as possible risk factors ($p < 0.1$) for postoperative pneumonia. After multivariable regression, independent association factors for postoperative pneumonia were identified: preoperative smoking (adjusted OR (aOR): 5.205, 95% CI [1.826–14.836], $p = 0.002$) and intraoperative hypotension (aOR: 3.082, 95% CI [1.447–6.432], $p = 0.003$) (Table 3).

### Secondary analyses

In the secondary analyses, we defined intraoperative hypotension as an intraoperative MAP decrease by > 30% from baseline and explored independent factors for postoperative pneumonia through multivariable analysis. Intraoperative MAP decrease by > 30% from baseline was not associated with postoperative pneumonia (aOR: 1.387, 95% CI [0.665–2.892], $p = 0.382$), while preoperative smoking (aOR: 4.253, 95% CI [1.525–11.855], $p = 0.006$) and duration of surgery > 4 h (aOR: 2.480, 95% CI [1.010–6.092], $p = 0.048$) were independently associated with postoperative pneumonia (Table 4).

### Exploratory analyses

As intraoperative MAP < 55 mmHg was independently associated with an increased risk of postoperative pneumonia in the present study, we further explored the association between the cumulative duration of intraoperative MAP < 55 mmHg and postoperative pneumonia through multivariable analysis (Table 5). Every 10 min of intraoperative MAP < 55 mmHg increased 2.618 times of the risk of postoperative pneumonia (aOR: 2.618, 95% CI [1.796–3.817], $p \leq 0.001$). We also explored the association between the cumulative duration of intraoperative MAP decrease by >30% with postoperative pneumonia. Every

**Table 1  Baseline parameters.**

|  | Without postoperative pneumonia, n (%) | With postoperative pneumonia, n (%) | *P* value |
|---|---|---|---|
| Age, (years) |  |  | 0.425 |
| <65 | 192 (82.4) | 41 (17.6) |  |
| ≥65 | 21 (91.3) | 2 (8.7) |  |
| Sex |  |  | 0.541 |
| Female | 144 (84.2) | 27 (15.8) |  |
| Male | 69 (81.2) | 16 (18.8) |  |
| BMI |  |  | 0.999 |
| >30 | 7 (87.5) | 1 (12.5) |  |
| ≤ 30 | 206 (83.1) | 42 (16.9) |  |
| Types of tumor |  |  | 0.811 |
| Meningioma | 114 (83.8) | 22 (16.2) |  |
| Glioma | 85 (83.3) | 17 (16.7) |  |
| Others | 14 (77.8) | 4 (22.2) |  |
| Preoperative hypoproteinaemia | 72 (83.7) | 14 (16.3) | 0.875 |
| Diabetes | 10 (83.3) | 2 (16.7) | 0.990 |
| Hypertension | 34(81.0) | 8(19.0) | 0.670 |
| Coronary artery disease | 2 (100.0) | 0 (0.0) | 0.999 |
| Anemia | 5 (71.4) | 2 (28.6) | 0.740 |
| COPD | 2 (100.0) | 0 (0.0) | 0.999 |
| Smoking history | 13 (61.9) | 8 (38.1) | 0.016 |

**Notes.**

BMI, body mass index; COPD, chronic obstructive pulmonary disease.

10 min of intraoperative MAP decreased by >30% from baseline did not increase the risk of postoperative pneumonia.

## DISCUSSION

In this single-center retrospective study, we observed a a history of preoperative smoking and intraoperative MAP < 55 mm Hg were independently associated with an increased risk of postoperative pneumonia in patients undergoing resection of intracranial tumors. Among the two identified risk factors, intraoperative hypotension might be one that can be actively addressed during surgery. Therefore, aggressive correction of hypotension during intracranial tumor surgery may play a crucial role in reducing the risk of postoperative pulmonary pneumonia in patients.

Intraoperative hypotension is common during general anesthesia (*Sessler et al., 2019*). If the blood pressure falls below the critical threshold required for adequate perfusion of vital organs, it may result in ischemic injuries (*Meng, 2021*). Several observational studies have shown that intraoperative hypotension is associated with postoperative complications, including acute kidney injury, myocardial injury after noncardiac surgery, and dysfunction of other vital organs (*Ruetzler, Khanna & Sessler, 2020*; *Shaw et al., 2022*; *Wijnberge et al., 2021*). Even one minute of intraoperative MAP < 55 mmHg significantly increased the risk

**Table 2** Perioperative parameters.

|  | Without postoperative pneumonia, n (%) | With postoperative pneumonia, n (%) | P value |
|---|---|---|---|
| Tidal volume (ml/kg) |  |  | 0.536 |
| >8 | 27 (87.1) | 4 (12.9) |  |
| ≤8 | 186 (82.7) | 39 (17.3) |  |
| Allogeneic blood infusion | 32 (86.5) | 5 (13.5) | 0.564 |
| Intraoperative MAP < 55 mmHg | 53 (72.6) | 20 (27.4) | 0.004 |
| Intraoperative MAP decreased > 30% from baseline | 128 (82.1) | 28 (17.9) | 0.538 |
| Neostigmine admission | 21 (87.5) | 3 (12.5) | 0.761 |
| ASA > 2 | 93 (78.8) | 25 (21.2) | 0.082 |
| Controlled ventilation > 4 h | 176 (80.7) | 42 (19.3) | 0.011 |
| Intraoperative fluid infusion > 2,000 ml | 154 (81.9) | 34 (18.1) | 0.359 |
| Estimated blood loss > 500 ml | 43 (84.3) | 8 (15.7) | 0.813 |
| Duration of surgery > 4 h | 119 (77.3) | 35 (22.7) | 0.002 |
| Body position |  |  | 0.634 |
| Supine position | 165 (83.8) | 32 (16.2) |  |
| Lateral position | 24 (77.4) | 7 (22.6) |  |
| Prone position | 24 (85.7) | 4 (14.3) |  |

Notes.

MAP, mean arterial pressure; ASA, American Society of Anesthesiologists.

**Table 3** Multivariable regression analysis for the association between postoperative pneumonia and intraoperative hypotension (MAP < 55 mmHg) alongside other potentially related factors.

|  | Adjusted OR | 95% CI | P value |
|---|---|---|---|
| Intraoperative MAP < 55 mmHg | 3.082 | 1.447 to 6.432 | 0.003 |
| Smoking history | 5.205 | 1.826 to 14.836 | 0.002 |
| Controlled ventilation > 4 h | 5.787 | 0.637 to 52.580 | 0.119 |
| Duration of surgery > 4 h) | 2.261 | 0.910 to 5.617 | 0.079 |
| ASA > 2 | 1.863 | 0.916 to 3.786 | 0.086 |

Notes.

OR, odds ratio; CI, confidence interval; MAP, mean arterial pressure; ASA, American Society of Anesthesiologists.

**Table 4** Multivariable regression analysis for the association between postoperative pneumonia and intraoperative hypotension (MAP decreased > 30% from baseline) alongside other potentially related factors.

|  | Adjusted OR | 95% CI | P value |
|---|---|---|---|
| Intraoperative MAP decreased > 30% from baseline | 1.387 | 0.665 to 2.892 | 0.382 |
| Smoking history | 4.253 | 1.525 to 11.855 | 0.006 |
| Controlled ventilation > 4 h | 4.783 | 0.548 to 41.740 | 0.157 |
| Duration of surgery > 4 h) | 2.480 | 1.010 to 6.092 | 0.048 |
| ASA > 2 | 1.870 | 0.932 to 3.751 | 0.078 |

Notes.

OR, odds ratio; CI, confidence interval; MAP, mean arterial pressure; ASA, American Society of Anesthesiologists.

**Table 5  Multivariable regression analysis for the association between postoperative pneumonia and intraoperative hypotension (every 10 min of MAP < 55 mmHg) alongside other potentially related factors.**

|  | Adjusted OR | 95% CI | *P* value |
|---|---|---|---|
| Intraoperative every 10 min of MAP < 55 mmHg | 2.618 | 1.796 to 3.817 | <0.001 |
| Smoking history | 5.903 | 1.959 to 17.787 | 0.002 |
| Controlled ventilation > 4 h | 4.515 | 0.495 to 41.151 | 0.181 |
| Duration of surgery > 4 h | 2.301 | 0.888 to 5.963 | 0.086 |
| ASA > 2 | 2.213 | 1.026 to 4.769 | 0.043 |

Notes.

OR, odds ratio; CI, confidence interval; MAP, mean arterial pressure; ASA, American Society of Anesthesiologists.

of postoperative cardiovascular complications (*Gregory et al., 2021*). However, research is scarce exploring the relationship between intraoperative hypotension and postoperative pneumonia in patients undergoing intracranial tumor resection.

Several studies have observed an association between intraoperative hypotension and postoperative infection. *Zhang et al. (2021)* retrospectively explored risk factors for postoperative infection in 880 patients who underwent gastrectomy for gastric cancer. Their results indicated that intraoperative SAP < 90 mmHg for > 10 min was associated with a higher incidence of postoperative infection (*Zhang et al., 2021*). Other studies also observed that intraoperative SAP < 80 mmHg increased postoperative wound infection in patients after major abdominal surgery (*Babazade et al., 2016*; *Yamamoto et al., 2007*). Although the exact mechanism behind the association of intraoperative hypotension with postoperative infections is currently unclear, it is speculated that the reduced tissue defense capabilities due to insufficient tissue perfusion because of hypotension may increase the risk of infection.

Currently, there is limited research on the relationship between intraoperative hypotension and postoperative pneumonia. We hypothesize that intraoperative hypotension may increase the risk of postoperative pneumonia through several potential mechanisms in the present study. Firstly, inadequate tissue perfusion during hypotension can lead to insufficient blood flow to vital organs such as the lungs, impairing oxygenation and increasing susceptibility to pulmonary infections. Secondly, ischemia-reperfusion injury caused by hypotension and subsequent blood pressure recovery may trigger inflammatory responses and oxidative stress, further damaging lung tissue. Additionally, hypotension and surgical stress can suppress immune function, weakening the body's defense against pathogens. Hypotension may also impair respiratory defense mechanisms by causing ischemia of the respiratory mucosa, disrupting ciliary movement and mucus clearance. Furthermore, prolonged recovery due to hypotension may increase the need for bed rest and mechanical ventilation, both of which are risk factors for pneumonia. Lastly, hypotension may induce systemic inflammatory response syndrome (SIRS), releasing inflammatory cytokines that further harm lung tissue. Therefore, intraoperative hypotension may indirectly elevate the risk of postoperative pneumonia through these interconnected pathways.

Relative definitions of intraoperative hypotension are widely applied in clinical practice (*Bijker et al., 2007*). In previous studies, the relationship between the definition of relative intraoperative hypotension and postoperative outcomes remained unclear. In a retrospective cohort study that included 18,756 surgical patients, Monk and colleagues observed that intraoperative MAP decreased > 50% from the preoperative level that lasted more than 5 min elevating the risk of mortality within 30 days after surgery, while intraoperative MAP decreased less than 50% from the preoperative level did not increase the risk of 30-day mortality (*Monk et al., 2015*). *Gregory et al. (2021)* retrospectively analyzed 621,482 intraoperative records. Their results suggested that intraoperative MAP decreased > 40% from baseline was not associated with major adverse cardiac or cerebrovascular events (*Gregory et al., 2021*). We observed that intraoperative MAP decreased > 30% from the preoperative baseline was not associated with postoperative pneumonia. Because of the paucity of studies on intraoperative hypotension and postoperative pneumonia, more research is needed to explore whether the use of a relative definition of intraoperative hypotension helps to improve patient prognosis.

At the same time, this study cannot confirm that hypotension is inherently harmful. Most current observations suggesting the detrimental effects of intraoperative hypotension are derived from retrospective studies (*D'Amico et al., 2023*; *D'Amico & Landoni, 2024*). Prospective randomized controlled trials and meta-analyses have failed to establish intraoperative hypotension as a direct cause of postoperative adverse complications. On the contrary, in certain patient populations, those in the hypotension group may even benefit. Additionally, intraoperative blood pressure may not accurately reflect tissue perfusion. Continuous infusion of catecholamines, while increasing blood pressure, can cause microcirculatory vasoconstriction, leading to inadequate tissue perfusion (*D'Amico et al., 2025b*). Therefore, during surgery, ensuring adequate blood supply to vital organs should not rely solely on maintaining blood pressure above a single threshold. Instead, it is essential to individualize the hypotension threshold based on the patient's specific condition, optimize cardiac function, and adjust fluid balance to ensure adequate perfusion of vital organs (*D'Amico & Landoni, 2024*).

This study has several limitations. Firstly, since we collected only adult patients who underwent surgery for intracranial tumors in a single center and returned directly to the ward, the representativeness of this study is limited. The positive association between intraoperative hypotension and postoperative pneumonia observed in this study requires a larger population to be examined. Secondly, due to the retrospective nature of the present study, other confounders that were not routinely recorded may confound our results. Perioperative lung-protective ventilation strategies such as intraoperative pulmonary resuscitation maneuvers are not routinely documented in the anesthesia information system. Furthermore, our anesthesia information system did not capture values of positive end-expiratory pressure (PEEP). Therefore, this study cannot exclude whether using pulmonary reanimation maneuvers or PEEP would have interfered with our results. Thirdly, details of preventive strategies for postoperative pneumonia that applied before and after surgery in surgical wards, such as deep-breathing exercises, were not routinely documented in our institution. In the present study, we cannot exclude the

effect of undocumented preventive strategies on our results. Future studies are needed to collect more complete postoperative data on postoperative pneumonia to clarify the etiology of postoperative pneumonia. Fourly, in this study, we did not explore the association of vasopressors or inotropic agents to postoperative pneumonia. The choice of medication to correct hypotension for each patient was determined by the attending anesthesiologist based on the patient's condition at the time. The medications used included norepinephrine, phenylephrine, ephedrine, dopamine, and others. Therefore, this study cannot use a uniform vasopressor or inotropic agent to analyze whether they are associated with postoperative pneumonia. Future prospective studies need to standardize the strategy for correcting intraoperative hypotension to investigate whether vasopressors or inotropic agents are related to postoperative pneumonia. Fifth, there are numerous methods available to quantify intraoperative hypotension, such as single-event duration, cumulative duration, area under the curve (AUC), time-weight average (TWA), among others. However, there is still insufficient evidence to determine which method of quantifying intraoperative hypotension best reflects the impact of intraoperative hypotension on postoperative complications. *Wesselink et al. (2018)* summarized recent clinical evidence on the relationship between intraoperative hypotension and postoperative complications. They observed that any exposure to MAP < 55 mmHg increased the risk of postoperative cardiovascular complications. Therefore, we selected whether patients experienced MAP < 55 mmHg during surgery to explore the relationship between hypotension and postoperative pneumonia in neurosurgical patients. Finally, in this retrospective study, the observed association between intraoperative hypotension and postoperative pneumonia cannot establish a causal relationship between the two, primarily due to factors such as confounding variables, unclear temporal sequence, selection bias, measurement errors, and the inability to determine biological mechanisms (*D'Amico et al., 2025a*). Retrospective studies can only suggest a potential correlation but cannot confirm a causal link. To establish causality, prospective studies or randomized controlled trials are typically required for further validation.

## CONCLUSIONS

Intraoperative MAP < 55 mmHg was associated with postoperative pneumonia in patients undergoing intracranial tumor resection. Our findings may provide clinically relevant information in designing prospective studies to evaluate possible etiology of postoperative pneumonia and could shed insights on intraoperative blood pressure management in neurosurgeries.

### Funding
The authors received no funding for this work.

### Competing Interests
The authors declare there are no competing interests.

## Author Contributions

- Yuan Chang conceived and designed the experiments, analyzed the data, prepared figures and/or tables, authored or reviewed drafts of the article, and approved the final draft.
- Yanqiong Wang performed the experiments, prepared figures and/or tables, and approved the final draft.
- Yinyan Zhou analyzed the data, prepared figures and/or tables, and approved the final draft.
- Huamei Chen performed the experiments, analyzed the data, prepared figures and/or tables, and approved the final draft.
- Yuanhua Li performed the experiments, prepared figures and/or tables, and approved the final draft.
- Ruhua Long analyzed the data, prepared figures and/or tables, and approved the final draft.
- Jianlin Shao conceived and designed the experiments, authored or reviewed drafts of the article, and approved the final draft.

## Ethical Approval

The following information was supplied relating to ethical approvals (*i.e.*, approving body and any reference numbers):

Research Ethical Committee of the First Affiliated Hospital of Kunming Medical University (2024-L-188, 2024-07-25).

## Data Availability

Raw data is available in the Supplemental Files.

## Supplemental Information

Supplemental information for this article can be found online at http://dx.doi.org/10.7717/peerj.19514#supplemental-information.

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
