# Peer review of "Intraoperative hypotension and postoperative pneumonia in patients after selective intracranial tumor resection: a retrospective cohort study"

_PeerJ, doi:10.7717/peerj.19514_

## Round 0.1 · original submission · Major Revisions

Please address all the comments of the reviewers. In particular, give due consideration to the comments from Reviewer 1

·

Basic reporting

The submission seems to be a mix of two different manuscripts. One being a systematic review and the other one being a retrospective study. At this point basic reporting cannot be assessed properly.

Experimental design

The submission seems to be a mix of two different manuscripts. One being a systematic review and the other one being a retrospective study. At this point experimental design cannot be assessed properly.

Validity of the findings

The submission seems to be a mix of two different manuscripts. One being a systematic review and the other one being a retrospective study. At this point validity of the findings cannot be assessed properly.

Reviewer 2 ·

Basic reporting

I would like to thank the authors for their work on this manuscript and for the opportunity to review it. This study investigates the association between intraoperative hypotension and the risk of postoperative pneumonia in adult patients undergoing intracranial tumor resection. By analyzing data from 341 patients over a five-year period, the authors identify key risk factors, including preoperative smoking history and intraoperative mean arterial pressure (MAP) < 55 mmHg, as significant contributors to postoperative pneumonia. The study is well-written and addresses an important clinical question. However, there are several points that require further clarification and discussion.

Experimental design

I do not have issue on experimental design

Validity of the findings

The background section suggests that intraoperative hypotension increases the risk of adverse events based on previous studies. However, to my knowledge, there are no randomized controlled trials demonstrating a causal relationship between intraoperative hypotension and worse outcomes. Observational studies can establish associations but are not sufficient to infer causation. It would be important for the authors to clarify this distinction and ensure that the discussion reflects the limitations of observational data in drawing causal conclusions.

The first sentence of the Discussion should clearly state the main finding of the study.

In the limitations section, the authors should acknowledge the intrinsic limitation of an observational study, specifically addressing the risk of observational interpretation fallacy ( DOI: 10.1111/jep.14288). Given the study design, causal relationships cannot be definitively established, and unmeasured confounders may still influence the results. Clarifying this limitation would strengthen the discussion and provide a more balanced interpretation of the findings.

There are two meta-analyses of randomized controlled trials indicating that the association between intraoperative hypotension and adverse outcomes observed in observational studies does not imply causation( DOI: 10.1016/j.bja.2023.08.026, DOI: 10.1097/CCM.0000000000006314). On the contrary, when patients are randomized to lower blood pressure targets, they often experience benefits. This has led to the concept of 'protective hemodynamics,' a strategy that accepts lower blood pressure targets as potentially advantageous (DOI: 10.1053/j.jvca.2024.10.021, DOI: 10.1097/MCC.0000000000001205) . These findings should be thoroughly discussed in the manuscript to provide a more comprehensive and balanced perspective on the topic.

·

Basic reporting

Thank you for inviting me to review this article. I’m confused that there were two abstracts, table 1 and table2. The abstract was not consistent with the manuscript. There may be significant error during the submission. I suggest to resubmit this article for further review.

Experimental design

For the abstract of “Postoperative hypotension may increase cardiovascular complications: a systematic review”, the results and the conclusion was contradictory.
For the manuscript “Intraoperative hypotension was associated with postoperative pneumonia in patients after selective intracranial tumor resection: a retrospective cohort study”.
In line 42-44: From this study, we just concluded that there was relationship between intraoperative hypotension and postoperative pneumonia, we can not conclude that Intraoperative hypotension was associated with an increased risk of postoperative pneumonia.
Line 89: The approval date should be indicated. The CONSERT checklist also should be provided.
Line 102: Since all patients were underwent general anesthesia, what dose the method of anesthesia mean?
Line 102-103: Protective pulmonary ventilation has significant impact on postoperative pulmonary complications, there has impact on the result of this study when intraoperative ventilation strategy were determined by attending anesthesiologists.
Line 106-108: Please provide the criteria of patients admitted to PACU or ICU after surgery.
Line 118: hypoproteinemia should be anemia?
Line 135-140: Should every patients undergo chest x-ray or CT examination if there has not symptoms of pneumonia?

Validity of the findings

Line 151-155: The figure 1 was incorrect?
Line 151-152: Patients were enrolled between April 1, 2015, and June 30, 2018. But in line 93-94, patients were enrolled between January 1, 2018, and December 31, 2022.
Table 2: Minute ventilation, should be tidal volume?
Table 2: The data unable to reflect the duration of hypotension, I suggest using more accurate methods to record hypotension, such as total duration of hypotension, area under the curve (AUC) of hypotension.
In results section: The intraoperative fluid infusion was associated with postoperative pneumonia, as well as the intraoperative inotropic drugs usage. More indicators should be included in this study.
In discussion section: The more updated literature should be cited, the relationship between hypotension and postoperative pneumonia should be discussed in detail.

Additional comments

This was a well designed study, hypotension affects organ perfusion, previous studies mainly focused on the relationship between acute kidney injury, myocardiac complicaitons, stroke, delirium and intraoperative hypotension, this study provided evidence of the associate between intraoperative hypotension and postoperative pneumonia.
In summary, this was retrospective study, some of the content seems to be another study, the authors should modify the submitted content. For the manuscript “Intraoperative hypotension was associated with postoperative pneumonia in patients after selective intracranial tumor resection: a retrospective cohort study”. More data which may leading postoperative pulmonary complications should be collected and analyzed.

---

## Round 0.2 · Minor Revisions

Thank you for addressing the reviewer comments. Could I please request that the following two points be addressed:

1. in the abstract, please add in the aOR results for prior smoking
2. in the methods section, please add in how you determined MAP. This has been addressed in the limitations section, but needs to be included in methods as well, as this is critical to understanding how you analysed and interpreted this data.

Reviewer 2 ·

Basic reporting

no comment

Experimental design

no comment

Validity of the findings

no comment

Additional comments

The authors have adequately addressed all my questions, and I believe the manuscript is now ready for publication.

·

Basic reporting

No comment.

Experimental design

No comment.

Validity of the findings

No comment.

Additional comments

This study investigated the relationship between intraoperative hypotension and postoperative pneumonia after intracranial tumor resection. The description was clear, the article structure was professional. The results had certain clinical significance.

---

## Round 0.3 · Minor Revisions

Per the previous discussion, this article needs to be a Research Article. In order to change that, please supply:

1. the raw anonymized data
2. a scan of the original document granting the ethical approval for your study
3. proof of waiver of the need for informed consent

---

## Round 0.4 · accepted · Accept

Thank you for uploading all of the requested documents. This manuscript is now ready for publication.